# Morphological Characteristics of a Horse Discovered in an Avar-Period Grave from Sâncraiu de Mureș (Alba County, Romania)

**DOI:** 10.3390/ani12040478

**Published:** 2022-02-15

**Authors:** Alexandru Gudea, Cristian Martonos, Călin Cosma, Florin Stan

**Affiliations:** 1Department of Anatomy, Faculty of Veterinary Medicine, 3–5 Manastrur Street, 400372 Cluj-Napoca, Romania; alexandru.gudea@usamvcluj.ro (A.G.); florin.stan@usamvcluj.ro (F.S.); 2Institute of Archaeology and Art History Cluj-Napoca, 12–14 M. Kogalniceanu Street, 400084 Cluj-Napoca, Romania; cosma2165@yahoo.com

**Keywords:** Avar, horse, archaeozoology

## Abstract

**Simple Summary:**

The present study deals with the morphological description of the animal remains in an Avar grave that contained the remnants of a horse burial, along with its owner, in a cemetery situated in Sâncraiu de Mures, Alba County, Romania, located geographically in the Transylvanian Plateau. The investigation of the grave and the data about the remains were published by archaeologists from the Institute of Archaeology of Cluj Napoca, Romania. For analysis and study, the animal bone remains were sent to the Faculty of Veterinary Medicine Cluj-Napoca. After a careful analysis of the morphological characteristics of the bones, the conclusions showed that the bones belonged to one young horse of 2–2.5 years, with an average recalculated shoulder height as being situated in the 1300–1400 mm interval. The study provides a set of morphological data regarding the characteristics of the horses from this scantily represented history in Romania, and the results show that the data obtained are consistent with the similar Avar horse burials in the region.

**Abstract:**

Archeozoological studies provide an insight into human–environment relations, bringing important information on the morphology of the animal and the role of the animal and its functions. The purpose of this study was to reveal the morphological characteristics of the horse identified in an 8th century BC Avar cemetery dated, as it resulted from the investigation carried out on the materials presented to the Anatomy Lab of Department of Comparative Anatomy. The cleaning and conditioning of the materials were performed in the lab, followed by anatomical and osteometrical study. The identification of the species, the osteometrical interpretation and the assessment of age at death constituted the basis of the main conclusions. Based on the morphological and metrical data, we concluded that the fragments originated from a single young horse individual (*Equus caballus*) no older than 2.5 (2–2.5 years age span) included into class 5 of height (Vitt scale) with slender extremities. An overview of the available data (horse morphology) from similar sites in Romania and neighboring territories (Hungary and Croatia) is presented, with the intention of a general framing for the characteristics of the horses used by the Avar populations.

## 1. Introduction

Archeozoology focuses on the study of the animal bones discovered in archaeological sites, aiming at providing a more complete insight into the relationships between humans and the environment, especially animals, thus discovering information about the animal species and their morphological characteristics [1]. A special feature of several human cultures in the ancient world was the ritual treatment of the horse. Such a treatment was spread all over Europe, in different populations such as the Prussians and other populations in the Balts [2,3]. The Avar populations that were spread all over Europe, mainly in the eastern and central parts of the continent, seemed to have practiced such ritual treatments as well [4]. The Avars (according to Chinese historical sources) were identified with the Juanjuan population living around Baikal Lake. This population was banished westbound around 555 PCh by Western Turks.

Some researchers assume that the Avars were the Hephthalites who crossed the Caucasus and started to move from the Caspian Sea towards Europe. According to sources, the Avars were the leading tribe that controlled and led with a firm hand the various ethnic groups and tribes that were migrating towards Europe [5]. The Avars conquered some territories from Central and South-Eastern Europe and, together with the North-Pontic regions, formed a political entity called the Avar Khanate. Its center was located in Pannonia. It was presumed that, by that time, ca 20,000 people, mostly warriors, entered Europe and remained there for about two centuries [5,6].

Several studies have pointed out the importance of the horse in the Avar communities. The ritual treatment of the horse seems to have been a common practice with the Avars. There is a large number of horse burials in the Carpathian Basin. They started to be classified into a typology [7,8]; the first list of the main types of burials appeared round 1962; later, the problem was largely debated by European professionals. As for the origin of horse burials in the “Avar milieu” of the Carpathian Basin, researchers (among them, Hungarian ones) [7] agree that this funerary practice was not a specific Avar custom [9]; it was adopted by the Avar warriors from the populations that they encountered on their way through Asia before reaching the Carpathian Basin [7,10,11].

Generally, the graves containing whole horse skeletons were men’s graves and belonged to the Avar military elite, an idea supported by the funerary finds as well. The horse was considered a life partner of the warrior. The reason why it had been buried along its owner was to serve its owner in the afterlife; this would be an explanation for the presence of harness parts and weapons in their common (owner and horse) grave.

In other cases, at the death of the warrior, the sacrificed horse was consumed by the entire community as part of the burial rituals. As a result, only some bones of the horse skeleton were placed in the grave. Most of these graves include skulls, mandibulae, horse teeth and bones from the lower (distal) part of the feet. The human graves that contain horses or only bones of the horse skeleton are not an exclusive attribute of the military elites. The presence of horses inside the graves rather signals the social status of the deceased. Not only ordinary warriors but also free and rich people, including women and children, have been buried with horses [7]. The Avar communities considered horses to be “magic”, possessing supernatural powers. Their death, natural or by sacrifice carried out by the community, was a significant event in the life of the community. The horse was then honoured with a lot of pomp when it was placed in the grave. The different types of horse burials from the Avar Khaganate of the Carpathian Basin, observed from one cemetery to another, from one geographic region to another, had an ethnic connotation, especially during the Early Avar period, when the relations between the different tribes that formed the Avar conglomerate were still incipient and very fragile [7]. When several types of horse burials meet in an Avar cemetery, it can be the case of cohabitation of two or more tribes with different ethnicities.

The persons buried with horses in the Gepid-Avar or Avar cemeteries from the Transylvanian Plateau and Charpatian Basin came from the ranks of the Avar warriors. The varied status of these warriors is mirrored by the richness and diversity of the funerary finds.

The person from Sâncrai buried with a horse was a warrior from the elite of the Avar community. This is evidenced by the belt with which the man was buried, adorned with appliques and bronze belt tongues. The warriors from Unirea 2/Veresmort, Șpălnaca and Cicău can be placed in the highest position of the Avar hierarchy in Transylvania [7]. The woman buried at Șpălnaca with her jewellery and costume accessories, who was accompanied by a horse, in its turn put inside with harness parts and accessories, certainly occupied a superior position within the Gepid-Avar community. According to the funerary finds, also, graves no. 9, 11, 42, 76, 85 and 133 from Bratei 3 were possibly women burials. The vast majority of the graves with horses accommodated adult persons. There are, however, several graves of adolescents identified in the cemetery at Bratei 3 (M. 11 and 133) [12] and of children, not very numerous (Band and Bistrița), buried together with horses or with bones of the horse skeleton, thus suggesting their upper status [7].

### The Archaeological Context

In between 2016 and 2021, a series of archaeological diggings were performed in the plane cemetery (dated eighth century) from Sâncrai Village, located in the Transylvanian Plateau. The site (Figure 1) is located in the area known as “La Cioca”, in between Sâncrai Village, Andrei Mureșan Street and Ciumbrud “Bokk-Pista”. The cemetery is located on the left bank of the Mureș River on the first terrace. One hundred and ten graves were investigated, dating mainly from the 8th century AD. From an ethnocultural point of view, the site belongs to the Avar culture of the Carpathian Basin [13]. In the spring of 2021, in the middle of a group of eight graves, a double human and equine tomb was discovered in a NW–SE orientation. After the initial stripping of the topsoil, at the −40-cm level, a back-colored elongated spot in the yellow clay soil was visible. At the depth of −100 cm, two tomb structures were identified. The diggings continued with a hand spatula until the elements of the tomb were revealed. The collection of the artifacts was done by hand, no sieving techniques being used. After the hand collection of the bones of horse and human, the soil under the bone layer was sampled for palynological analysis.

Inside the tomb, the deceased was laid in the grave in the anatomical position, with the head NW and feet SE. The human was buried with a very rich belt and many fittings. Close to it, separated by a 10-cm tall curb, an articulated horse skeleton was identified, placed on its right side with feet tucked under its belly (Figure 2).

The orientation of the horse remains was with head placed SW and tail NW. The horse appeared to be fitted with harness pieces (bit, mouthpiece and stirrup), while parts of weapons were also discovered close to the skeleton (spear point). All the elements of this archaeological inventory point to the fact that the deceased might have been a high-ranking Avar warrior [13].

The purpose of this study is to provide elements regarding the morphology of the horse for this very little studied period in the history of Romania and to attempt a framework, based on other available bibliographical sources, for the general morphology of the species in the light of a wider territorial perspective.

## 2. Materials and Methods

The bony material was presented to the Anatomy Lab of the Faculty of Veterinary Medicine Cluj-Napoca Romania, as it was roughly harvested from the excavation site. The material was cleared, washed and gently cleaned from all debris in the Anatomy lab.

The pieces of the skeleton were very little affected by disarticulation and dispersal processes, but diagenetic fossilization and later mechanical alterations (including exposure to soil moisture and acidity) made the investigation more difficult than usual, as the bones were frail and prone to further fragmenting [14,15].

The bones were left to dry naturally for a certain amount of time; after which, a general assessment was made to establish the direction for the identification. As the bones were intensely affected by the exposure to physical agents (soil acidity), they called for a consolidation procedure. A construction product based on PVA (polyvinyl acetate) was used in the primary approach in an attempt to preserve and consolidate the frail elements and to connect the extremities of the long bone fragments [16,17].

Another drying phase followed, and a proper analysis was initiated by evaluating the degree of fragmentation and the potential of rebuilding the separated parts of the bones.

After the reconstruction of the long bones mainly (epiphyses and diaphysis), the unidentifiable pieces were counted and separated from the identifiable material. The specific identification was made [18,19,20,21,22,23] and the osteometric evaluation was performed based on the standard measurements [24]. Evaluation of the epiphyseal stages was made based on the standard sources [18,23,25,26].

## 3. Results

### 3.1. The Archeozoological Investigation

The analyzed sample is attributed to one single species: the horse (Equus caballus). The distribution of the bone fragments is illustrated in Table 1.

#### 3.1.1. Scapula

Fragments from both scapulae were identified, but only the right one was measurable at the level of the articular angle (Table 2, Figure 3). The supraglenoid tubercle (tuberculum supraglenoidale) was ossified, which is an indicator of an age over 10–12 months [26].

#### 3.1.2. Humerus

The stylopodial fragments were preserved in a decent condition. None of the pieces were ossified proximally (Figure 3). The measurements were possible only in the case of the left humerus, the right one being very degraded at the level of the proximal processes.

As the ossification process was not fully closed at the proximal extremity, we concluded that the animal’s age at the time of death was situated in the 15–18 months to 42 months interval [26]. The osteometrical available data (Table 3) allowed us the recalculation of the height based on the lateral length [27]. In our basic calculations, the recalculated height was 1266 mm.

#### 3.1.3. Radius and Ulna

Both bones were decently preserved and were undamaged (Figure 3). The presence of the epiphyseal line is to be mentioned in the case of both distal radial fragments. Similarly, the tuber olecrani lacked ossification in both bones. These data indicated again the age of death situated in the interval of 15–18 months to 42 months [26]. The height estimation was made on the calculations (Table 4) using Kiessewalter’s factor [25] and turned a value of 1388 and 1401 mm

#### 3.1.4. Carpal Bones

A series of (almost) complete carpals were identified (accessory carpal bone right and left, ulnar carpal bone: left, intermediate carpal bone: right and left, radial carpal bone- right and left, and os carpale I and os carpale II), with no significant data to be subtracted from their study. No cutmarks or other specific interventions that might suggest usage of the skeletal elements for other purposes [28] were noted on the bones of this anatomical segment.

#### 3.1.5. Metacarpals

The well-preserved bones (Figure 5) allowed us to give a precise morphological diagnosis and the estimation of the age of death. The bones are fully ossified, suggesting that the time of death for the investigated individual was over 15 months [26]. The bones also allowed us a complete series of measurements (Table 5) that permitted a height and gracility estimation.

In accordance with the formula suggested by Boessneck, the recalculated height, based on the lateral length of the metacarpals, is situated in the interval of 139–141 cm (class 5 on Vitt’s scale) [26,29] with a gracility index, in accordance with the Brauner or Cersky scales [26], pointing to an individual with slender limbs (upper limit: 14.2–14.4).

#### 3.1.6. Pelvis

Three identifiable fragments of pelvic bone were studied: 1one fragment of iliac body from the right part; one acetabular fragment from the right part and another iliac, ischiadic and acetabular part from the left. The pieces are heavily degraded, and no measurements were taken. The only indicator for age estimation is represented by the acetabular area (over 12 months of age) [26].

#### 3.1.7. Femur

Heavily degraded (Figure 4), the femur provides just limited morphological details. One can notice the fact that the proximal elements of the epiphysis are not fused and ossified (as noted for the right femur), while the left one lacks the elements of the proximal extremity. The distal epiphyses are unfused but retrievable in both studied specimens. The estimated age, based on the fusion stages, indicate an individual that could have been younger than 36 months at the time of death. The recalculation of the height (with a high amount of reserve, due to degradation-approximated Glc (length at articular head) of 326 mm) is indicating a height of 1144 mm [30].

#### 3.1.8. Tibia

The pelvic zeugopodial fragments were preserved in a decent state. The proximal epiphyses are unfused for both studied specimens (right and left), while the distal epiphyses are fused (Figure 4). This is an indicator of the age over 24 but under 42 months [26].

The metrical data available (Table 6) led to the recalculation of the height in accordance with the Kiesewalter formula [25]. The values are situated between 146 and 148 cm, including the individual in the fifth class of heights, in accordance with Vitt’s scale [19].

#### 3.1.9. Tarsals

From the tarsal series, several pieces were identified: talus (os tarsi tibiale), calcaneus (os tarsi fibulare), cuboideum (os tarsale quatrum) and os tarsi centrale (Table 7 and Table 8). The tuberosity of calcaneus (os tarsi fibulare) was clearly unossified, which is an indicator of an estimated age below 36 months [26]. No cutmarks were visible [28]. The state of preservation is quite poor.

#### 3.1.10. Metatarsals

As seen on the metacarpals, the closure of the epiphyseal plates indicates again the same age span for the time of death for this individual: over 15 months [16] (Figure 5). The height recalculation was made based on the formulas of Boessneck [15] and provided values between 1369 and 1375 mm (Table 9). According to Vitt’s scale, the individual can be framed in class 5 of heights [19].

#### 3.1.11. Phalanges

The distal parts of the limbs were much better preserved than the proximal ones, allowing us descriptions and study of the proximal and middle phalanges. All of the first and middle phalanges were ossified, which is an indicator of an individual over 12–15 months [16]. The metrical data is summarized in the following table (Table 10). There was only one fragment of a distal phalanx identified, but highly degraded, so no measurements were possible.

#### 3.1.12. Ribs

Nine proximal parts of the os costale were retrieved. No important data were obtained from their study, as the pieces were fragmented and affected by environmental factors.

#### 3.1.13. Vertebrae

The atlas and axis were identified, but no significant data could be extracted, as most of the pieces were fragmented and degraded. A series of thoracic vertebrae were also identifiable, as their bodies were unossified, a clear indicator for ages below 4–4.5 years [26]. The other 10 fragments were not attributed to any of the vertebral divisions, as they were represented by spinous processes or smaller articular fragments of the body. We must mention the fact that there is no evidence of pathological or subpathological changes in the area of the spine (as an indicator for riding and load bearing), as suggested by literature-sources [21,31,32,33], but this might be explained by the relatively early age of the identified individual.

##### 3.1.14. Cranium

This part of the skeleton is highly fragmented, so only a few morphometrical data could be collected from a relatively large number of bony fragments.

The mandibular parts (47 fragments, including dentition) allowed a series of estimations regarding the age but provided very little metric data. A complete series of disparate permanent incisors with no wear whatsoever made us assume the fact that these are unerupted teeth, with a logical conclusion of the fact that the individual was younger than 2.5 years [8,18]. Another series of not surely identified disparate molars and premolar fragments led to no firm conclusions, but some implanted lacteal molars (with not a certain allotment) suggest once again an age of more than 1 year, maybe even 2 years of age.

The fragments that could reconstruct the posterior part of the left mandibular ramus and the body of the mandible (including the articular extremity) show the existence of the first molar (which is an indicator of age over 15 months [18]).

The fragments of the left mandibular ramus show similar features, except for most of the dental elements that could not be evaluated due to high degree of fragmentation. The only assessment that we could perform was due to the presence of the unerupted last molar (indicating an age below 3.5 years [18]).

The other fragments are parts of the maxillae mainly, with a clear display of the facial crest and parts of the zygomatic bone down to the alveolar margin of the maxillae. Here, we could evaluate and observe the existence of the first two erupted permanent molars, as the third molar was still unerupted. This is a clear indicator of age over 2 years but not more than 3–3.5 years [18].

The only assumption we make, based on the observation of the fragmented dentition, for the assessment of the gender, in this case, is that the fragment probably originated from a female individual, but this statement is to be regarded with reserve, as the absence of the canine tooth is not a certain indicator of a female in respect to sexual dimorphism [23,26,34].

### 3.2. Contextualization of the Osteometric Data

The present study brings into light, to the best of our knowledge, a new series of elements referring to the morphology of the horse from the Avar period, along with other two or three studies about the same period for the area of our country. This is the reason why the addition of these data is an important contribution to the knowledge for the period in terms of animal morphology that results from this study.

Based on the morphological and metrical data, we can conclude that the studied fragments originated from a single young horse individual (Equus caballus). The individual was no older than 2.5 years, presumably with an age situated in the 2–2.5 years age span, with no record of pathological changes to the skeletal elements

The multitude of metrical data helped us estimate the shoulder height for our individual that was situated in the interval of 1350–1400 mm, values that indicate the lower limit for the group of medium-sized horses [35]. The metrical data we obtained were calculated from the measurements of different long bones in correlation with different formulas for the calculation of the height.

One must mention the fact that some of the bones were not fully ossified, a thing that leaves some room for interpretation with respect to the degree of the body development of a young individual such as the one we are invoking here.

Based on the modern domestic animal biology and breeding information, the body size of a 2-year-old yearling will be 96% of the mature weight and body development [25], thus leaving us to assume that the combined method calculation of the height (that involved the GL (greatest length) of the humerus, radius and metacarpals with a correction due to an inclined axis of the stylopodium, as suggested by Bartosiewicz [31]), provided quite accurate final results.

With this formula, a final value of 1300 mm was the result. With the added correction of another 4 to 5%, one can assume a value of 1360 mm based on this calculation method.

The other estimations that were based on the simple average data provided by the individual calculations of the height of all the long appendicular elements provided an average value of 1344 mm (±25).

Both calculations came up with quite similar values, so we could consider both reliable, as far as metrical data are concerned. This includes the identified individual in class 5, in accordance with Vitt’s scale [29], with slender extremities (a value that is situated, in accordance with the metacarpal data at the upper limit for slender-legged individuals but not falling into the medium-slender category) [21,26,36].

## 4. Discussion

The data provided by this study added some interesting morphological information to the scantily researched subject on the morphology of the Avar period horses.

Up to the present day, very few scientific works tackling the subject have been published in Romania—the Avar horse from Veresmort (Unirea II), Alba County and from Săliște-Cicău, Alba County [37,38]; more morphological data are available from a series of archaeological studies on the territory of Hungary [4,35] and Croatia [21]. All these three regions have published studies targeting similar historical periods and populations (the Avars) and geographical proximity (all three Romanian sites are in the northwestern part of Romania, which is bordered by Hungary eastbound, while the Croatian sites are located southwest from the Hungarian Border).

A general comparison of the estimated heights (Figure 6) from all these sites shows a quite narrow interval for the medium heights of the studied individuals. It is worth noticing the variability of data from the Hungarian sites (where the samples were rich, with a significant number of individuals described), but, as stated in the analyzed papers, most of the individuals were slender or medium slender-legged. This is proven by the diagram of the height distribution, which illustrates the aforementioned aspects.

We also must mention the fact that, in the case of available data from Cicău and Veresmort [26,27], we performed the recalculation of all the heights based on the GLs (greatest lengths) of all the long bones available in the published study (as average values) and applied the formula of Bartosiewicz [21] that makes the angular correction for the humerus. The latter provides an average value slightly smaller than the one mentioned by the authors (−50 mm).

To illustrate some morphological features, we used the metrical data of the metapodials (that seem to be the most relevant) for the morphological characterization of the horse individuals.

As we can notice from the dispersion chart for the metacarpals (pointing to the GL versus diaphyseal width) (Figure 7), the data is quite consistent with the sources we consulted [4,21,35]. One can notice the placement on the lower registry for our individual, close to the values recorded in some Hungarian Avar sites. Another representation that uses the GL vs. diaphyseal index (as an indicator of gracility, as mentioned in many sources) (Figure 8) shows quite a similar fact: placement in the lower-left registry, indicating a gracile and not—so-tall individual, similar to the ones visible in some Hungarian sites.

For metatarsals, a similar representation of the GL vs. diaphyseal width (Figure 9) shows again a similar placement for our data: a less tall and gracile individual, whose values are blended with the other data from Hungarian sites in the lower registry of our chart. The Gl vs. diaphyseal index (a more accurate representation) (Figure 10) indicates a similar positioning in the lower and gracile side of the graph, similar with some of the Hungarian sites.

## 5. Conclusions

Based on the slender extremities and the average value for the reconstructed withers height (with all subjective factors related to biological variability and ossification stage of this individual), our study yielded results that are in favor of classification of the animal into the classically so-called “Eastern type” of animals (that were smaller in size and gracile). This classically admitted morphotype is mentioned, starting from the time of the Iron Age as coexisting with the “Western type” (a larger and stouter morphological type, e.g., Celtic horses) on the territory of Central and Eastern Europe, a group that was constantly “infused” with horses that migrated with the successive wave of migratory populations [35]. This method of framing seems to be nowadays related more to morphological characteristics rather than geographical landmarks or territories. It is well-known that the entire historical period is marked by ample movements of the human populations and, as history recorded, quite significant heterogeneity in terms of horse morphology and genetics [35,39,40].

Another accepted (yet somehow arbitrary) framing for horses, in this case, might be the distinction of “elite” from “ordinary” types of horses that indicates a threshold of 1400-mm shoulder height [26], designating our individual as an “ordinary” one placed at the upper limit of this class. The situation points to the class of rouncies and destriers (as mentioned mainly for medieval samples) [41], clearly focusing, in fact, on the physical qualities a horse might have.

In a very general perspective, the literature agrees on using the terms of “war horse” and “ordinary horse” pointing to the main usage of the animal rather than indicating a conformational or morphological type. Judging after the evidence shown by the archaeological context, naming this individual a war horse is not a mistake, being part of the ritualic warrior’s mound that was sacrificed at the time of burial [4], with no indication of another functional use of the animal except for the utilitarian one [28].

The results of the archeozoological investigation carried out on the horse identified in the warrior’s tomb from Sâncrai are important from the perspective of seventh to eighth century history of Transylvania, being one of the few scientific archeozoological attempts for the area of the Transylvanian Plateau. This study, along other archaeological finds, proves the fact that the Avar Warrior that reached this territory in his journey from the Eurasian Steppe at the end of the seventh century, ruling out a series of hypotheses that state differently (the Northern European theory or the Western Carpathian Basin). For the Transylvanian region, there are few evidence for this type of man and horse joint burials. This is the reason why this studied tomb confirms the typical Avar ritual of the personal horse being placed alongside its owner at the moment of death.

## Figures and Tables

**Figure 1 animals-12-00478-f001:**
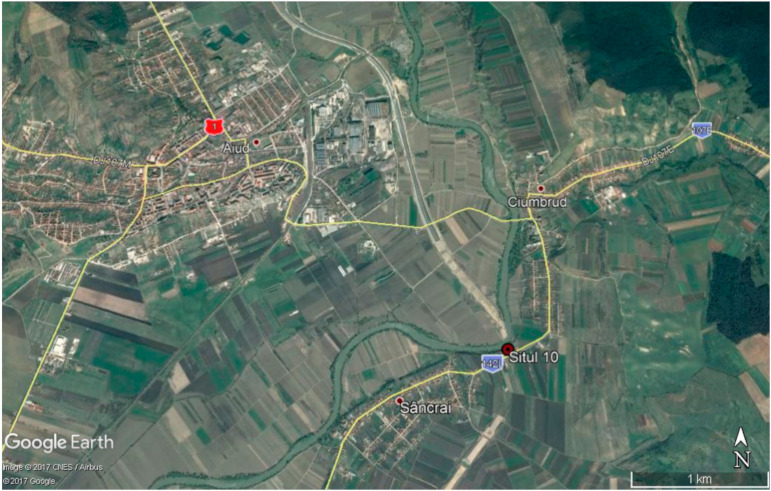
Location of the archaeological site and proximity of Aiud (Alba County, Romania) Google sites capture.

**Figure 2 animals-12-00478-f002:**
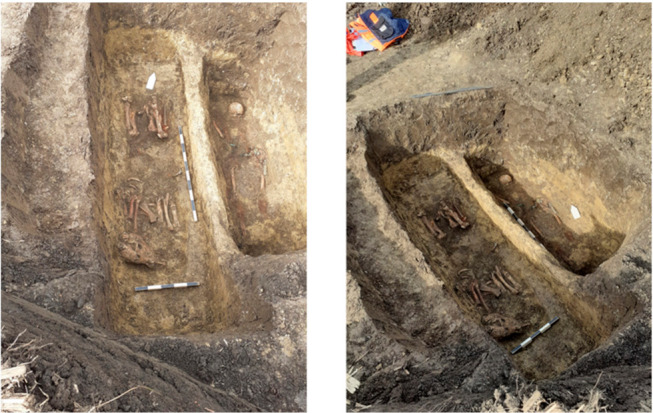
Archaeological diggings: Sâncraiu site and the double tomb.

**Figure 3 animals-12-00478-f003:**
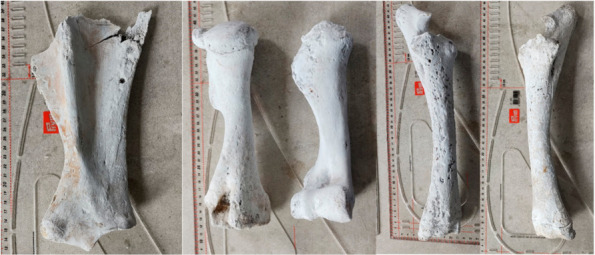
Hardened and reconstructed horse bones: scapula, humerus, radius and ulna.

**Figure 4 animals-12-00478-f004:**
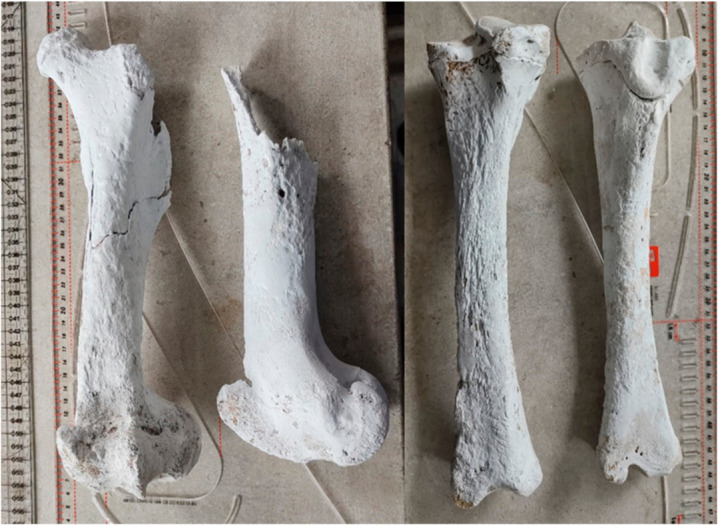
Hardened and reconstructed horse bones: femur and tibia.

**Figure 5 animals-12-00478-f005:**
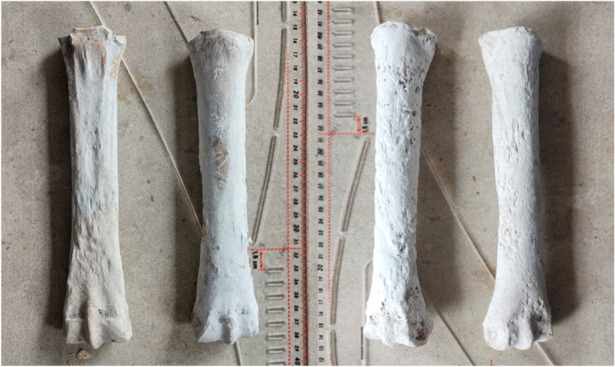
Hardened and reconstructed horse bones: metacarpals (**left**) and metatarsals (**right**).

**Figure 6 animals-12-00478-f006:**
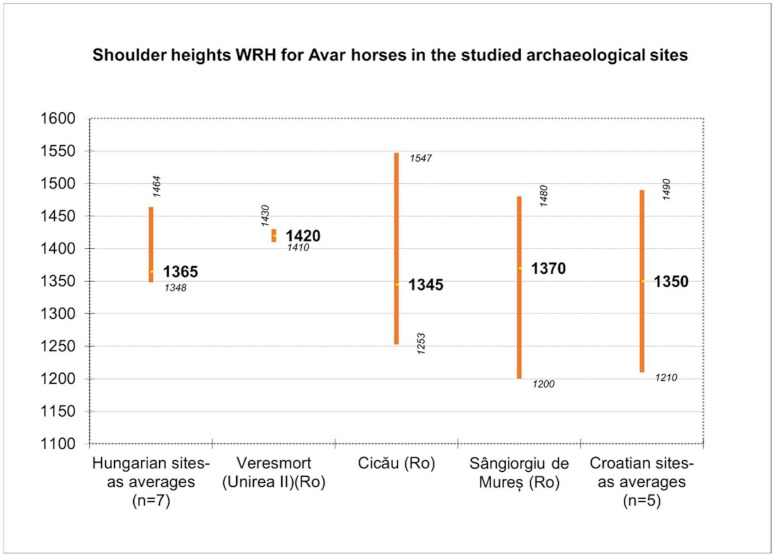
Shoulder height for Avar horses (recalculated or average values expressed in cm, with minimal and maximal values).

**Figure 7 animals-12-00478-f007:**
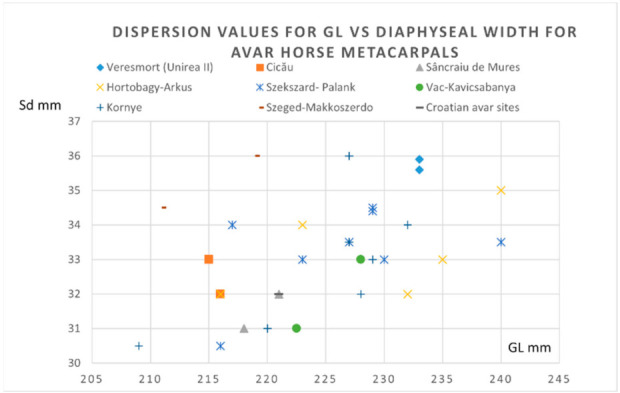
Metacarpals- dispersion chart for GL (greatest length) vs. Sd (minimal diaphyseal width) for Avar horse finds.

**Figure 8 animals-12-00478-f008:**
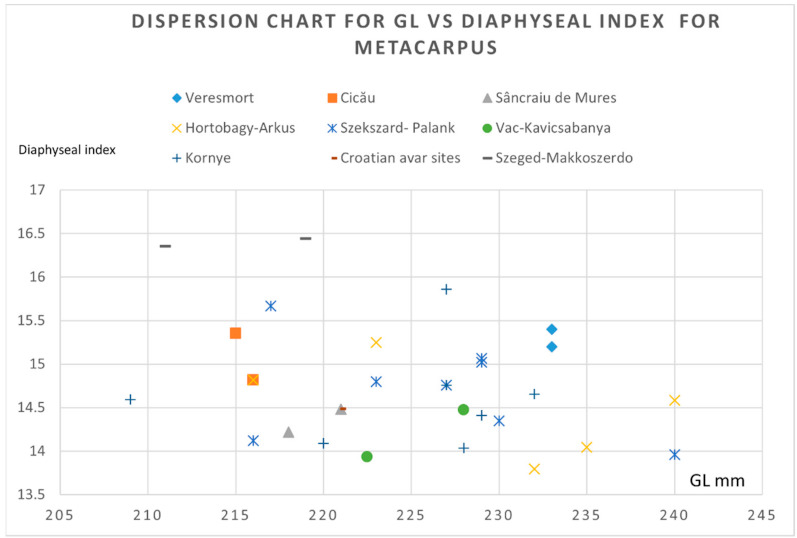
Metacarpals: dispersion chart for GL (greatest length) vs. the diaphyseal index (sd/GL × 100) for the Avar horse finds.

**Figure 9 animals-12-00478-f009:**
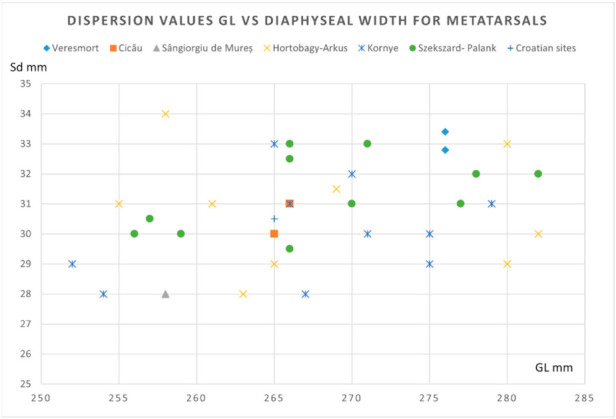
Metatarsals: dispersion chart for GL (greatest length) vs. Sd (minimal diaphyseal width) for the Avar horse finds.

**Figure 10 animals-12-00478-f010:**
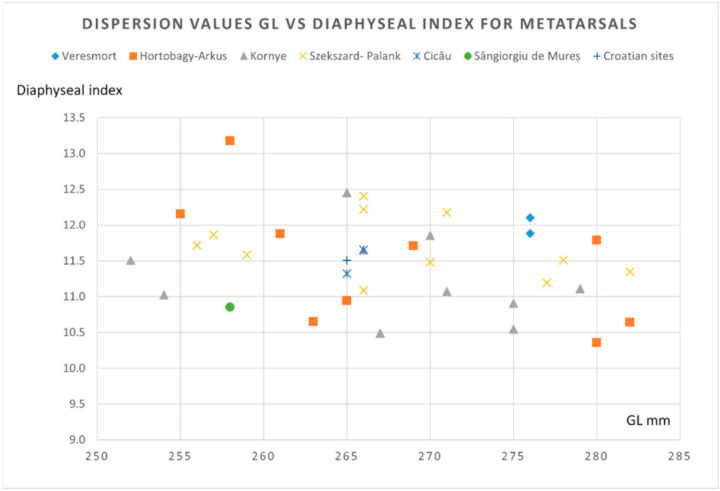
Metatarsals: dispersion chart for GL vs. the diaphyseal index (sd/GL × 100) for the Avar horse finds.

**Table 1 animals-12-00478-t001:** Distribution of the skeletal components.

Bone	Right	Left
Scapula		1
Tibia	1	1
Metacarpal III	1	1
Metatarsal III	1	1
Humerus	1	1
Femur	1	1
Radius and ulna	1	1
Pelvis	1 acetabular, 1 iliac body	
Os costale	9 proximal extermities and shafts	
Vertebrae	1 atlas, 1 axis, 10 thoracic fragments, 10 unattributed fragments	
Phalanges 1	4	
Phalanges 2	4	
Phalanges 3	1	
Patella	1	
Tarsals	2 talus (os tarsi tibiale), 2 calcaneus (os tarsi fibulare), 1 cuboideum (os tarsale IV), 1 os tarsi centrale	
Carpals	2 os carpi accessorium, 1 os carpi ulnare, 2 os carpale III, 1 os carpale IV	
Splanchnocranium	68 fragments of different sizes, mainly with dentition	
Neurocranium	47 fragments	
Dentes	8 incisors, 6 lacteal molars	
Unidentified	38	

**Table 2 animals-12-00478-t002:** Measurements for the scapula (mm): (6) LG (greatest length), (7) BG (anteroposterior glenoidal diameter), (4) SLC (breadth of the articular angle) and (5) GLP (anteroposterior diameter of the articular process).

Side	6	7	4	5
R	51	42	60	87
L	-	-	-	-

**Table 3 animals-12-00478-t003:** Humerus measurements (mm): (1) GL (greatest length), (2) GLI (lateral length), (3) GLC (maximum length at head), (4) Bp (proximal width), (5) Dp (anteroposterior width), (6) Sd (minimal diaphyseal width), (7) Bd (distal width) and (8) Bt (width of the trochlea).

Side	1	2	3	4	5	6	7	8
R						32	40	66
L		260 *	247	75	88	32	39	66

* reconstructed length.

**Table 4 animals-12-00478-t004:** Radius and ulna measurements (mm): (11) GL (greatest length), (15) Bfp (diameter of the proximal articular facet), (18) Bd (distal width), (16) Sd (minimal diaphyseal width) and (2) Gl1 (maximal external length).

Side	11	15	18	16	2
R	323	76	67	34	384
L	320	74	66	35	390

**Table 5 animals-12-00478-t005:** Metacarpus measurements (mm): (1) GL (greatest length), (2) Gl1 (maximal external length), (3) Ll (external length), (4) Bp (proximal width), (5) Dp (anteroposterior width), (9) Bd (distal width), (10) Dd (minimal anteroposterior diaphyseal diameter) and (6) Sd (minimal diaphyseal width).

Side	1	2	3	4	5	9	10	6
R	218	214	212	47	31	45	31	31
L	221	217	214	47	32	46	34	32

**Table 6 animals-12-00478-t006:** Tibia measurements (mm): (1) GL (greatest length), (2) Ll (external length), (3) Bp (proximal width) and (6) Bd (distal width).

Side	1	2	3	6
R	340	320	86	65
L	335	315	88	66

**Table 7 animals-12-00478-t007:** Calcaneus (os tarsi fibulare) measurements (mm): (1) GL (maximum length) and (2) Gl (greatest breadth).

Side	1	2
R	105	48
L	105	42

**Table 8 animals-12-00478-t008:** Talus measurements (mm): (1) GL (maximum length) and (4) L lat tr (external length of the trochlea).

Side	1	4
R	54	57
L	54	58

**Table 9 animals-12-00478-t009:** Metatarsus measurements (mm): (1) GL (greatest length), (2) Gl1 (maximal external length), (3) Ll (external length), (4) Bp (proximal width), (5) Dp (anteroposterior width), (9) Bd (distal width), (10) Dd (minimal anteroposterior diaphyseal diameter) and (6) Sd (minimal diaphyseal width).

Side	1	2	3	4	5	9	10	6
R	258	255	254	45	36	41	34	28
L	257	253	251	44	37	44	34	28

**Table 10 animals-12-00478-t010:** Phalanges measurements (mm): (1) GL (greatest length), (2) Bp (proximal width), (6) Bd (distal width) and, (5) Sd (minimal diaphyseal width).

	F1	F1	F1	F1	F2	F2	F2	F2
1	77	74	80	81	45	45	45	44
2	46	44	48	46	44	46	45	44
6	37	37	39	39	41	40	39	41
5	30	30	32	33				

## Data Availability

The data that support the findings of this study are available from the corresponding author, A.G., upon reasonable request.

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
