# Peer review of "Morphological Characteristics of a Horse Discovered in an Avar-Period Grave from Sâncraiu de Mureș (Alba County, Romania)"

_animals, 2022, doi:10.3390/ani12040478_

Round 1
Reviewer 1 Report
The relation between human and their beast of burden, from the beginning of equid domestication until today, are a fascinating subject of research. Many studies discussed the regions of domestication, where others analyzed the taxonomy of domesticated horses. This work includes a good morphometric review of an ancient horse (possibly female) that was buried beside her owner (a warrior). At the same time, this work lacks an analysis of the relevant cultural context, and the human-animal relations in the avar period. Which types of connection existed between human and horses, and why warriors were buried beside horses in this society.
The goal of the work is to provide a detailed analysis of the horse remains, yet it is important to reveal also some information about the human buried beside it. Human-horse relations in ancient Romania are fascinating, important and therefore worth publication.
Yet, although the work is important, several points require further detailing and focus before this work can be published, as follows:
1) It is expected to refer to the conclusions of the study with an appropriate modesty. For example, I'd suggest removing the word "complete" in:
'Archaeozoological studies provide a complete insight into the human-environment relations'
2) It is important to clarify specific sentences which are unclear for all readers: who are the Avars? What is the Avar period? Why rituals were developed around horses (this was suggested both in the introduction and discussion but does not reviewed).
3) Since the remains of humans, buried beside horses, were found in additional sites in the Avar period, it is expected to provide a larger perspective of human-horse relationship in this period.
4) Methods – it is important to indicate the collection methods – were they based on sediment sifting or not. In the article, only the methods of archeological material transfer, conservation and identification are mentioned.
5) Results – the illustration and legends are well done.
6) conclusions- the morphometric and aging analysis are excellent, yet the article lacks any insight on the context of human-horse burial, and a hypothesis on why this ritual existed.
Author Response
Response to Reviewer 1 Comments
The relation between human and their beast of burden, from the beginning of equid domestication until today, are a fascinating subject of research. Many studies discussed the regions of domestication, where others analyzed the taxonomy of domesticated horses. This work includes a good morphometric review of an ancient horse (possibly female) that was buried beside her owner (a warrior). At the same time, this work lacks an analysis of the relevant cultural context, and the human-animal relations in the avar period. Which types of connection existed between human and horses, and why warriors were buried beside horses in this society.
A significant section dealing with general framing of the cultural context and the relation human-animal has been inserted in the Introductory part. This section is pointing to the topics mentioned in the reviewer’s comment, creating a general, wider perspective on these issues (role of the horse, practice of horse burials, typology, distribution of these rituals across Europe).
The goal of the work is to provide a detailed analysis of the horse remains, yet it is important to reveal also some information about the human buried beside it. Human-horse relations in ancient Romania are fascinating, important and therefore worth publication.
As data from the archaeological diggings were already published, we made a short inventory of the humans buried along the horses in the area of our interest. The description and elements are placed also in the introductory part, with reference to Veresmort, Cicau and Sancrai (sites that were used also for the comparative data in terms of horse morphology), Spalanca, Sugud
Yet, although the work is important, several points require further detailing and focus before this work can be published, as follows:1) It is expected to refer to the conclusions of the study with an appropriate modesty. For example, I'd suggest removing the word "complete" in:
'Archaeozoological studies provide a complete insight into the human-environment relations'
removed
2) It is important to clarify specific sentences which are unclear for all readers: who are the Avars? What is the Avar period? Why rituals were developed around horses (this was suggested both in the introduction and discussion but does not reviewed).
A small subsection with data on the avars and historical framing of the expansion of the populations originating from Baikal Lake throughout Europe and the settlement in the Pannonian area has been implemented into the Introduction section.
3) Since the remains of humans, buried beside horses, were found in additional sites in the Avar period, it is expected to provide a larger perspective of human-horse relationship in this period.
The newly introduced subsection in Introduction tackles this human-horse relationship from a broader perspective
4) Methods – it is important to indicate the collection methods – were they based on sediment sifting or not. In the article, only the methods of archeological material transfer, conservation and identification are mentioned.
A brief description of the archaeological procedures was inserted in a subchapter called “The archaeological context”
5) Results – the illustration and legends are well done.
Thank you
6) conclusions- the morphometric and aging analysis are excellent, yet the article lacks any insight on the context of human-horse burial, and a hypothesis on why this ritual existed.
Thanks to all your previous notes and suggestions, by the insertion of a much broader description of the historical context, the importance of the horse exploitation and data regarding the archaeological investigation itself, I hope we succeeded in reaching this goal.
Reviewer 2 Report
The article ‘Morphological characteristics of a horse discovered in an avar-period grave from Sancraiu de Mures (Alba County, Romania)’ presents an osteological investigation of a buried horse with the aim to shed light on horse morphologies in the area and time period under study.
Overall, it is a good paper that is worth publishing because of the scarce data there is so far in Romania regarding horse morphology during the Late Antiquity and beyond. Also, it contributes to the discussion of some trends that have been suggested for other areas of Europe. However, there are some things that should be reviewed before accepting it for publication. I would be willing to review a revised version if the following weaknesses pointed are improved:
- Materials and Methods:
- Giving more details about the horse’s burial would be necessary. Were the bones articulated? Was the skeleton recovered complete? How many fragments were recovered?
- Results:
- A taphonomic section before showing anatomical representation would be suitable. The authors explain (lines 74 and 182) that environmental agents affected the bones, but they do not give more details about this exposure.
- Has any pathology been documented on the bones?
- The authors explain that no cut marks were documented on the phalanges (line 120). What about on the other bones?
- Discussion
- This section needs to be rewritten. I would suggest adding most of the current discussion to the Results section in a new subsection that could be called: 3.1. Contextualization of the osteometric data/ 3.1. Horse morphology in a broad context or something similar (from line 256 onwards), in which to show the osteometric comparison of authors’ data with other available data.
In the Discussion section the authors should discuss and interpret their data presented in the Results section, and also the results obtained from the comparison with the other available data from the area and time period under study. This information is missing in the current manuscript.
- The authors should explain and discuss in the Discussion section what they call in the Conclusions section “eastern group of animals” and also “elite and ordinary types of horse”.
- English needs some editing. There are two spelling mistakes in the title.
Specific comments:
- I would suggest presenting in one single table all the osteometric data.
- Measurements of X and Y-axis of Figures 7, 8, 9, and 10 must be specified in the figures and also in the figure legends. What measurement are the authors referring to when they write dyphisis? They must also explain how they have calculated the “dyphiseal index”.
Author Response
The article ‘Morphological characteristics of a horse discovered in an avar-period grave from Sancraiu de Mures (Alba County, Romania)’ presents an osteological investigation of a buried horse with the aim to shed light on horse morphologies in the area and time period under study.
Overall, it is a good paper that is worth publishing because of the scarce data there is so far in Romania regarding horse morphology during the Late Antiquity and beyond. Also, it contributes to the discussion of some trends that have been suggested for other areas of Europe.
Thank you. Thanks for your patience in through roughly checking of our paper.
However, there are some things that should be reviewed before accepting it for publication. I would be willing to review a revised version if the following weaknesses pointed are improved:
- Materials and Methods:- Giving more details about the horse’s burial would be necessary. Were the bones articulated? Was the skeleton recovered complete? How many fragments were recovered?
This correction was made on the Introduction section, in a newly introduced section describing more precisely the entire archaeological context. The overall number of fragments-as counted on the archeozoological investigation- is mentioned in the Results section.
There is also a more ample framing of the historical context and short mentioning about the cultural context and the relation human-animal.
- Results:
- A taphonomic section before showing anatomical representation would be suitable. The authors explain (lines 74 and 182) that environmental agents affected the bones, but they do not give more details about this exposure.
As suggested,there is a new paragraph that tackles the changes and processes that characterize the sample in respect to the typical taphonomic stages inserted in the Materials and methods” section.
- Has any pathology been documented on the bones?
No. No significant pathological changes were recorded on the studied material. This fact is mentioned now in the Discussion part, first paragraph.
- The authors explain that no cut marks were documented on the phalanges (line 120). What about on the other bones?
No other visible cutmarks were recorded on the examined material
- Discussion
- This section needs to be rewritten. I would suggest adding most of the current discussion to the Results section in a new subsection that could be called: 3.1. Contextualization of the osteometric data/ 3.1. Horse morphology in a broad context or something similar (from line 256 onwards), in which to show the osteometric comparison of authors’ data with other available data.
In the Discussion section the authors should discuss and interpret their data presented in the Results section, and also the results obtained from the comparison with the other available data from the area and time period under study. This information is missing in the current manuscript.
We have made the restructuring of the Discussion section and the Results section by adding the suggested subchapter and moving the content into the higher section. The remaining second part was, as suggested, part of the Discussion section.
- The authors should explain and discuss in the Discussion section what they call in the Conclusions section “eastern group of animals” and also “elite and ordinary types of horse”.
The distinction of two important horse groups is considered customary as the classical archaeozoological state. One of the most important sources- S. Bokony (Bokony 1974) has used this classification starting from the Iron Age Period in Europe. He mentions the division of the two groups- the eastern and western one designating horses with slightly different morphological features existing (based on osteological evidence) on the actual territory of Central and Eastern Europe. From these two main groups, for example, the Scythian horses (as representatives of Eastern Group) and Celtic horses (Western Group) are to be framed in later periods and several so-called domestication centers appeared based on these two populational groups. The roman military horse seems to be originating from horses of Western Group- Scytian horses. Cited diversified breeds are also mentioned by the same sources pointing to Syrian and Thracia Horses that belonged to Eastern Group, The migration Period in Europe is mentioned as being characterized for Germanic and Celtic people with horses belonging to the western group while Avars, Huns seem to be related with horses of Eastern Group. This is also the period where the running “warm blooded” and trotting“cold blooded” are mentioned and the connection between the Avar horses is made with the running horses type (stated by another classical archeozoologist- J.U.Duerst in the early 1900!), Overall, the historical period (with Germanic, Avar, Hungarian groups- mentioned by S.Bokony) seems to be characterized by the “significant influence of mass of eastern (group) horses that spread to Europe with the various successive waves of migration of peoples”. There is not a final agreement on this clear separation of morphotypes, but literature and scholars continue even nowadays this distinction in the attempt of the general framing the horse populations for these historical periods.
The elite and ordinary types are nothing but sort of extension of this division at a much smaller scale. Due to heterogeneity in population, it is admitted that a sort of empirical selection (regardless of many populational morphotypes) may lead to a sort of preference for a slightly larger, stouter set of individuals- called “elite horses” while other individuals, smaller, slender etc. might have been considered a “minus variant” not so desirable for breeding, not used for training and battles but for household usage.
- English needs some editing. There are two spelling mistakes in the title.
Corrected, spelling check redone and another grammar and proofing from a professional
Specific comments:
- I would suggest presenting in one single table all the osteometric data.
Unfortunatetly (although we felt the same way sometimes for the “economy of the page”) as the osteometrical elements are most of the times different from one anatomical sector to another and sometimes we do not even use the whole set of standard measurements for a specific bone, the headers of the tables are different (sometimes significantly different from bone to bone). This makes the fusion of tables impossible from editing perspective. Also, for the ease of usage for other zooarchaeologists, tables may stay separated as data within tables are much easier to use and extract.
- Measurements of X and Y-axis of Figures 7, 8, 9, and 10 must be specified in the figures and also in the figure legends.
The suggested changes were made. Axis legends were embedded into the graphs and explanatory notes were made in the figure legends
What measurement are the authors referring to when they write diaphysis? They must also explain how they have calculated the “dyphiseal index”.
Most probably a typo. The diaphyseal index is a standard osteometrical calculation made on long bones to express the ratio between the total length of the bone and the minimum width of the bone, thus expressing the slenderness of the bone. The calculation is explained in the legends for figures 8 and 10 too.
Reviewer 3 Report
This paper presents the metric analysis of a horse from a joint human-horse burial. While the zooarchaeological methods and approaches are sound, the authors fall short of justifying the significance of this single find for the broader readership of the ANIMALS journal.
Below are some specific thoughts for the manuscript:
Lines 222-223…this is the start of the discussion and it is unclear what the authors mean here by saying they have brought to light “a new series of elements referring to the morphology of the horse from the avar period…”. Up to this point, the authors have only reported the metric analysis of their single horse in line with previously published metrics, and have yet to introduce any new information besides the description of their horse. It is unclear why horses were important to the Avar period, or why this period is poorly understood from Romania. In many instances the authors appear to assume generalised familiarity with the long-term archaeological history of Romania, and leave out important contextual information which would emphasis the significance of their find.
Lines 228-231 - this exact information, or variations of it, have been repeated multiple times throughout the manuscript already. More careful editing could help to decrease instances of repetition.
For all figures, each figure needs its X and Y axis labelled with what data it is showing, and to have the units of measurement included. Throughout the manuscript authors report shoulder height in mm…but in some of the figures they appear to be showing cm, though the axis are unlabelled. these should be consistent.
For some figures, (Fig 6 for instance) because the authors are reporting averages, they also need to include sample sizes for each group.
In many instances, the appearance of the figures does not coincide with the discussion that relates to them. For example, in paragraph beginning Line 258: the discussion of additional data comes much later than the figures which it relates to. This leave the reader questioning the context when the figures are presented, and then having to backtrack once the context is finally provided much later on. Furthermore, insufficient data is provided on where the additional horse metrics have come from. Were they collected by the authors, or from previously published sources? How many sites are they, how many samples from each group? Where the same measurements taken in each case? There is also no contextual information on why Hungarian horses are an appropriate comparison here. Again the reader is left feeling like the authors have assumed a level of specialised knowledge that we do not have, and cannot expect from the broad readership of the journal.
The authors measure shoulder height from bones which have not fully ossified. They mention this could cause inaccuracies in their interpretations but do not indicate which bones are adult and which are estimates, potentially introducing bias into their interpretation of the data,.
Line 245 the authors switch to using cm here to report shoulder height when previously they had been using mm.
Throughout the authors reference to the “sources they consulted”, however these interpretations are only included as a reference, leaving the reader to rely on their own previous knowledge or having to go to each source to read those articles in order to interpret the validity of the claims and comparisons the authors are making.
Line 295 - this is the first mention of the “eastern group” and it’s not clear why or how this is a significant finding.
Line 297 - this is not “well know” to the interdisciplinary readership of the journal.
The conclusions present brand new information for the first time and no where is there a robust discussion of the ideas presented here. Mentioned for the first time in the conclusion are “easter group” of horses, “warhorses”, “elite” horses, and distinguishing between utilitarian and elite horses. None of this is explained or interrogated in the discussion section. Nor is the claim that this particular horse is a warhorse justified or supported by any of the evidence the authors provide, besides the archaeological context of the burial, which was apparent before any of the osteological analyses was conducted.
Overall, the authors present two pieces of novel information about a single horse skeleton - how tall the horse is, and how slender it is. However, they fail to impart why this is significant in the context of Avar horse, or other horses from this region or similar time period. Indeed, they provide almost no substantial context for the importance of this individual horse, and instead rely heavily on outside sources, which they reference but do not describe in enough detail for the reader to understand the significance of these results without reading each referenced paper themselves. While there is no need to repeat the findings of previous articles, enough information needs to be provided in this manuscript to that the relevance of the citations to the new data is clear. At this stage, this paper does not stand alone as an independent piece of work .
Given one of the aims of the Animals journal is to publish “ articles that provide an understanding of animals within a larger context “ this papers significant lack of context for an interdisciplinary readership is a disservice. While (zoo)archaeologists or specialists in the region of study may have enough background to interpret and appreciate this manuscript, the broader readership of ANIMALS which includes “zoology, ethnozoology, animal science, animal ethics and animal welfare” will struggle to decipher the relevance as it is currently presented.
The entire manuscript could use to be carefully proof read. Below is an incomplete list of grammatical errors:
“Avar” should be capitalised throughout
Line 10 - full stop missing at end of …Alba County, Romania
Line27 - ...’assessment for age at death they all stayed as foundation for the main conclusions’…possibly meant to be two sentences?
Line 30 - ‘hose morphology’ should read ‘horse morphology’
Line 43 - ‘archaeological digs’
Line 60-61 ‘high ranking’
Line 73 ‘ dry naturally’ not ‘dry naturally off’
Line 109 - unclear of meaning of “both bones were decently preserved and were integer(?)”
Line 236 “long bones”
Author Response
This paper presents the metric analysis of a horse from a joint human-horse burial. While the zooarchaeological methods and approaches are sound, the authors fall short of justifying the significance of this single find for the broader readership of the ANIMALS journal.
Thanks for your patience in through roughly checking of our paper. Your notes on the paper served as a good point for the improvement.
Below are some specific thoughts for the manuscript:
Lines 222-223…this is the start of the discussion and it is unclear what the authors mean here by saying they have brought to light “a new series of elements referring to the morphology of the horse from the avar period…”. Up to this point, the authors have only reported the metric analysis of their single horse in line with previously published metrics, and have yet to introduce any new information besides the description of their horse. It is unclear why horses were important to the Avar period, or why this period is poorly understood from Romania. In many instances the authors appear to assume generalised familiarity with the long-term archaeological history of Romania, and leave out important contextual information which would emphasis the significance of their find.
As a result of your note on the structure of the paper, a change has been made by splitting the part of the description of the finds, by the addition of a subchapter that presents first the result of the specific zooarchaeological investigation, followed by a clear dividing section that deals with a contextualization of our data the finds in the light of data from other investigated similar sites (Results section).
The importance of the horse and the historical frame was highlighted by a separate section that contextualizes all these elements, emphasizing the importance of such finds even from the introductory part.
As data is so scarce for the period, with few finds and even fewer archeozoological investigations done on discovered horses, we may consider that any piece of new info is adding knowledge to this little studied type of material.
Lines 228-231 - this exact information, or variations of it, have been repeated multiple times throughout the manuscript already. More careful editing could help to decrease instances of repetition.
As the Results part has been divided into 2 subsections, the information presented appears as a summing up of the information that results from the previous section. We tried to reduce the redundant information from this section.
For all figures, each figure needs its X and Y axis labelled with what data it is showing, and to have the units of measurement included.
Marked and edited
Throughout the manuscript authors report shoulder height in mm…but in some of the figures they appear to be showing cm, though the axis are unlabelled. these should be consistent.
We have changed the units in mm throughout all the text. All tables were referred as measurements in mm.
For some figures, (Fig 6 for instance) because the authors are reporting averages, they also need to include sample sizes for each group.
Clear reference into the corresponding text/graphs was inserted
In many instances, the appearance of the figures does not coincide with the discussion that relates to them. For example, in paragraph beginning Line 258: the discussion of additional data comes much later than the figures which it relates to. This leave the reader questioning the context when the figures are presented, and then having to backtrack once the context is finally provided much later on.
We think it is just a matter of “fit into page” issue that we hope to fix or improve in the corrected and reedited version, by arranging the graphs closer to their reference into text. Some of the graphs may have several references into the text but again the points of reference in different pages may make this correlation a bit difficult. We also do not know how the editor will shrink or enlarge the actual graphs, making the arrangement much better than our initial editing.
Furthermore, insufficient data is provided on where the additional horse metrics have come from. Were they collected by the authors, or from previously published sources? How many sites are they, how many samples from each group? Where the same measurements taken in each case? There is also no contextual information on why Hungarian horses are an appropriate comparison here. Again the reader is left feeling like the authors have assumed a level of specialised knowledge that we do not have, and cannot expect from the broad readership of the journal.
The metrical data from other sources were used as were published in raw-osteometric data tables (similar as the ones we have revealed in this study). This is quite a common procedure in osteometry, by using standard osteometrical sets of measurements (Desee or Boessneck) so each scholar can reuse or maybe reinterpret the data in the light, maybe, of a different approach (eg. as we did with raw osteometrical data from Cicau- that needed a recalculation based of an later-published formula). Another example is the raw osteometrical data from Croatian sites (where we used another set of measurements presented somehow differently- thus the use of averages presented by authors). The sample size was listed in the graph (as earlier suggested) for the Hungarian and Croatian sites, while the Romanian sites were about one single individual, separately published.
The choice for Hungarian and Croatian reference data is simple- we are comparing similar populations- as historical sources and evidence state- Avars that were spread on a large territory, but quite close geographically speaking with the area from Romania that is in discussion (northwestern part, bordered by the nowadays Hungary). A short note has been introduced in the text mentioning this geographical proximity and populational similarity. Such a thing is also mentioned in th newly- introduced section in the first part of the paper that frames the general historic context, few facts about the Avar population and the relationship between man and horse and the importance of the horse burial rituals.
The authors measure shoulder height from bones which have not fully ossified. They mention this could cause inaccuracies in their interpretations but do not indicate which bones are adult and which are estimates, potentially introducing bias into their interpretation of the data
This is a good observation. Based on a whole complete set of data, as bones belonged to one single individual, the potential bias induced by the unossified bone taken in consideration is reduced to a minimum. One must note the fact the final calculation for shoulder height estimation is based on a complex formula that takes into consideration several long bones and, in accordance to age estimation there was a extra correction added for the very final figure of the shoulderheight.
The shoulder height (re)calculation is called an “estimation” and it is not to be regarded as an absolute figure (such as many data in biology)
Line 245 the authors switch to using cm here to report shoulder height when previously they had been using mm.
Changes were made, as suggested by your previous comment.
Throughout the authors reference to the “sources they consulted”, however these interpretations are only included as a reference, leaving the reader to rely on their own previous knowledge or having to go to each source to read those articles in order to interpret the validity of the claims and comparisons the authors are making.
We may think that is a a citation procedure. There are spots in the main text that we reference as “consulted sources” with no direct referral to a citation, but, as we saw it, repetition of the sources (cited at the beginning of the section) may have seem redundant. We have inserted references also to the point indicated by our reviewer.
Line 295 - this is the first mention of the “eastern group” and it’s not clear why or how this is a significant finding.
This aspect was approached a little more as suggestions indicated. The eastern and western groups of horses are cited as the main populational groups that spread throughout Europe from the Iron Age onwards. It is a commonly accepted theory among archaeozoologists (stated early 1900 by J Duerst and confirmed and improved by S.Bokonyi) and explains the morphotypes and their variations among several groups of horse populations in a much later period, up to Migration Periods and even later.
Some brief mentioning were inserted in the Conclusion parts to clarify this theory, especially for a lower level of specialized knowledge.
Line 297 - this is not “well know” to the interdisciplinary readership of the journal.
The newly introduced historical frame section in the introductory part makes a bridge between the general framing, historical context etc and this last- mentioned fact referring to populational migrations of the era.
The conclusions present brand new information for the first time and no where is there a robust discussion of the ideas presented here. Mentioned for the first time in the conclusion are “easter group” of horses, “warhorses”, “elite” horses, and distinguishing between utilitarian and elite horses. None of this is explained or interrogated in the discussion section. Nor is the claim that this particular horse is a warhorse justified or supported by any of the evidence the authors provide, besides the archaeological context of the burial, which was apparent before any of the osteological analyses was conducted.
The previous comments of the reviewer and the newly introduced notes on history, explanations on the contextual situation on the horse populations may serve as good basis for interpretation. It is clear that on the basis of one single individual, such a framing is not recommended, but our approach tackles series of theories and currents that more or less target the same morphological difference- either a local one, as a result of individual biological diversity being the mainbase for empirical selection or a more general one, attempting to use classical scales or frames mentioned by classic sources.
Overall, the authors present two pieces of novel information about a single horse skeleton - how tall the horse is, and how slender it is. However, they fail to impart why this is significant in the context of Avar horse, or other horses from this region or similar time period. Indeed, they provide almost no substantial context for the importance of this individual horse, and instead rely heavily on outside sources, which they reference but do not describe in enough detail for the reader to understand the significance of these results without reading each referenced paper themselves. While there is no need to repeat the findings of previous articles, enough information needs to be provided in this manuscript to that the relevance of the citations to the new data is clear. At this stage, this paper does not stand alone as an independent piece of work .
The presentation of the morphological data is a novelty standalone. As long as there is just a few available similar elements in the literature, the addition of other morphological elements may serv as a step forward into the overall assessment of the animal population from a biological perspective, integrated into tha big frame of an historical era. The comparison with the little information that is available up to the date is nothing but an attempt on gathering information and processing data into a contextual frame.
Given one of the aims of the Animals journal is to publish “ articles that provide an understanding of animals within a larger context “ this papers significant lack of context for an interdisciplinary readership is a disservice. While (zoo)archaeologists or specialists in the region of study may have enough background to interpret and appreciate this manuscript, the broader readership of ANIMALS which includes “zoology, ethnozoology, animal science, animal ethics and animal welfare” will struggle to decipher the relevance as it is currently presented.
One of the main purposes of the study is to bring morphological data into light. We were somehow fortunate to have some reference data in the same territory and in the neighbouring geographical areas so there might be a beginning for a more ample framing. Such an attempt will be possible when for a territory, more similar morphological investigations will be available, but given the historical context and scarcity of finds up to date, this is a significant step forward in the assessment of horse in this region and the integration of these morphological data into the larger historical context of the Avar world onto the actual Romanian territory.
The entire manuscript could use to be carefully proof read. Below is an incomplete list of grammatical errors:
Another English proofing and checking have been performed and the modified material was again checked by a specialized English Language professional.
“Avar” should be capitalised throughout
Capitalized all
Line 10 - full stop missing at end of …Alba County, Romania
corrected
Line27 - ...’assessment for age at death they all stayed as foundation for the main conclusions’…possibly meant to be two sentences?
Rephrased and shortened sentence
Line 30 - ‘hose morphology’ should read ‘horse morphology’
corrected
Line 43 - ‘archaeological digs’
corrected
Line 60-61 ‘high ranking’
corrected
Line 73 ‘ dry naturally’ not ‘dry naturally off’
corrected
Line 109 - unclear of meaning of “both bones were decently preserved and were integer(?)”
corrected
Line 236 “long bones”
corrected
Reviewer 4 Report
Dear Authors,
Thank You for Your important work. I appreciate Your interest in the horse’s archaeozoology. Even that work is important and Your findings will for sure find the wide answer In archaeozoological world I must suggest the major revision (see remarks in pdf file) and English language improvement. Moreover I would like to ask You for your results discussing with works of Daugnora from Lithuania.
Reviewer

Author Response
Thank You for Your important work. I appreciate Your interest in the horse’s archaeozoology. Even that work is important, and Your findings will for sure find the wide answer In archaeozoological world I must suggest the major revision (see remarks in pdf file)
Thanks for your patience in through roughly checking of our paper. All suggestions from your pdf file were taken in consideration and corrections were made (typos, addition of measurements names in the tables, capitalizations etc). Suggestions regarding phrasal topics and expressions were also reconsidered and corrected. Many notes for the presentation on tables, usage of NAV and anatomical terms and mistakes were corrected too.
We saw that you made a note on the usage of the term “zootechnical” (Discussions) but we would like to keep it, as it points exactly to the animal breeding/husbandry in a wider perspective as a branch of animal sciences.
But and English language improvement.
Another English proofing and checking have been performed and the modified material was again checked by a specialized English Language professional.
Moreover, I would like to ask You for your results discussing with works of Daugnora from Lithuania.
At the time of the material elaboration, we were aware of the studies of scholars from Lithuania and one of the most important works regarding horses (Fages et al 2019) was in our focus, but as it did not target the historical era that we were studying, we did not mention it initially. We have also made a small notice on this work too, in the general context of the complexity horse genetics in the Conclusions part.
According to your suggestion, another works that tackle a sort of similar historic period, but another population (Bertasius-Daugnora 2001, Daugnora-Thomas 2002) were mentioned in our paper as a very good source and analogy in respect to human-horse burials in another geographical region (Introduction part).
Round 2
Reviewer 1 Report
The article has been corrected as per the requirements and no further corrections are needed.
Author Response
Thank you for your time and comments on our paper. They were really helpful!
Kind regards,
Reviewer 3 Report
I am satisfied with the extensive edits made to this manuscript, in particular those which now emphasis the significant context of the horse examine here, making the manuscript much more appropriate for a broad interdisciplinary readership of this journal.
Author Response
Thank you for your time and useful comments!
Respectfully,
Reviewer 4 Report
Dear Authors,
I appreciate Your work in manuscript improvement. Majority of my remarks were taken under consideration and introduced in text. Term “zootechnical’ is still controversial for me. I understand Your point of view, but be so kind and take under consideration other form. Maybe “modern domestic animals biology and breeding” will be better. I leave final decision for You.
Some small mistakes I highlighted and commented in pdf. file.
The quality of some figures (diagrams) must be improved. They are not sharp.
I recommend to accept the reviewed manuscript after minor revision.

Author Response
Thank you very much for your attention and kind comments. We have made the small Latin syntax changes and also changed the term zootechnical to the one suggested...
As far as images are concerned, I saw your file and there is a striking difference. In the word file the images are crystal-clear..... probably in the conversion to the pdf file something was lost. Most probably the editors will figure it out...if necessary I will provide the raw tiff files for them.
Kind regards from Romania,